# Fear of missing out, social media influencers, and the social, psychological and financial wellbeing of young consumers

**Abbey Bartosiak[1], Jung Eun Lee[2], Cäzilia Loibl[1]***

**1** Department of Human Sciences, The Ohio State University, Columbus, Ohio, United States of America,
**2** Department of Consumer and Design Sciences, Auburn University, Auburn, Alabama, United States of America

* loibl.3@osu.edu

## Abstract

This study examines the social, psychological, and financial wellbeing of younger consumers who follow social media influencers. The consumer panel MTurk was used to collect a U.S. sample of 863 adults aged between 18 and 40 who follow social media influencers. Structural equation modeling is used for data analysis. Greater fear of missing out is directly linked to lower levels of social, psychological, and financial wellbeing, however it is associated with stronger parasocial interactions with social media influencers. Parasocial interactions are, in turn, associated with higher levels of social, psychological, and financial wellbeing. This study is among the first to document the direct association of fear of missing out with lower social, psychological, and financial wellbeing and shows that parasocial interactions with social media influencers are a key pathway through which social media use can benefit the wellbeing of younger consumers.

## Introduction

An influencer is a person who has gained notoriety through social media and not through traditional celebrity means, like being a movie star or professional athlete. Social media influencer marketing involves partnerships between companies and influencers to endorse products and services. This strategy leverages the influence of individuals to promote brands and content to their social media followers, and it has emerged as a pivotal factor in consumer purchasing decisions [1,2]. This relatively new form of marketing has dramatically increased in popularity in the United States. Its market capitalization more than doubled from $6.5 billion in 2019 to 13.8 billion in 2021 and the number of firms using influencer marketing almost doubled from 4,000 in 2019 to 7,300 in 2021, with Instagram being the preferred channel [3]. A key reason for the success of social media influencer marketing is that followers feel they connect to the influencer like a friend [4]. Influencers define themselves as ordinary people so their product reviews are perceived as more authentic by followers than other forms of marketing [5]. Due to these perceived similarities, followers tend to develop emotional relationships with influencers when interacting with influencers' content on social media.

The emotional attachment and friend-like feeling associated with influencer interactions assimilate a parasocial interaction (PSI), which can be particularly attractive to consumers

**Data availability statement:** The SPSS data file is available from the OSF.IO database (URL: https://osf.io/v5zw6/?view_only=3fb68e-f318e14d069aa2531de96652f0).

**Funding:** The authors acknowledge generous funding from the Coca-Cola Critical Difference for Women Grant for Research on Women, Gender, and Gender Equity from The Women's Place at The Ohio State University. The funders had no role in study design, data collection and analysis, decision to publish, or preparation of the manuscript.

**Competing interests:** The authors have declared that no competing interests exist.

who have greater fear of missing out (FoMO) [6,7]. FoMO describes individuals who are preoccupied with what others are doing and experiencing [8]. Although recent studies have examined FoMO in the social media context [7], the relationship between FoMO and PSIs with social media influencers has received limited attention. This is surprising because, just as in real-life relationships, individuals with greater FoMO might benefit from relationships built with influencers if influencers are able to fulfill a consumer's need to stay connected and preoccupied with events happening in an influencer's life [6,7]. Bridging the gap of understanding the relationship between FoMO and PSIs with social media influencers will support further foundational research related to a number of areas, including social media influencer related consumption, sociopsychological needs and motivations of consumer spending habits, and needs gratification from consumerism [9].

Another gap in understanding how consumers experience social media influencer engagement lies in the link of PSIs with social media influencers and consumers' FoMO, which may influence their social, psychological, and financial wellbeing. A small number of research studies have shown that relationships with social media influencers differ in quality from interactions with friends and family on general social networking sites [10,11]. However, the pathways through which consumers interact with social media influencers, i.e., through FoMO, PSIs and social shopping, and how these pathways relate to social, psychological, and financial wellbeing is little understood. Our study contributes to bettering the understanding of behaviors related to individuals' wellbeing with respect to the interaction with social media influencers. Specifically, we explore how individuals' psychological anxiety (i.e., FoMO) can be mitigated through their sense of connection with influencers via PSIs, leading positive perceptions towards their wellbeing. This investigation is important due to the abundance of research on social media influencers focusing on their marketing effectiveness and the mimicry behavior they inspire [12], which often leads to negative consequences rather than positive effects on the social and psychological health of followers.

The focus of the current study is on younger consumers, age 18 to 40, who follow social media influencers. Framed through PSI, we investigate whether younger consumers with higher levels of FoMO are more likely to have stronger PSIs with social media influencers, and how these interactions relate to social shopping intentions as well as financial, social, and psychological wellbeing. This study will provide new insights into benefits and burdens of social media use and results in recommendations for interventions aimed at informing younger consumers about responsible social media practices.

## Conceptual framework

### Parasocial interaction (PSI) with social media influencer

To identify the influencer-follower relationship of younger consumers who follows social media influencers, we propose the PSI framework [13,14]. PSI refers to the psychological relationship developed by an individual after repeated exposure to a media personality [15]. In PSIs, individuals develop a relationship with personas while knowing that the relationship is not reciprocated by the media personality [13,14].

Research on PSIs began in the 1950s when mass media and television personalities began to gain traction in popular culture [15]. Recent research has transferred the well-established PSI framework to the social media context by examining the relationships and interactions between social media influencers and their followers [16]. Just like early literature that connected individuals to television personalities, the current literature connects consumers to social media influencers based on the same one-sided experience of a PSI. Despite the medium, the central theme has been carried forward: the media personality, in this case

the social media influencer, does not reciprocate feelings to an individual who follows and connects with them [17]. The perceived up close and personal relationship that social media influencers have with consumers through social media has allowed brands to capitalize on for promoting products and services effectively [18].

## Fear of missing out (FoMO)

FoMO is a personality trait that can be defined as a "pervasive apprehension that others might be having rewarding experiences from which one is absent" [8, p. 1841]. FoMO is often related to individuals' anxieties and worries regarding the missing out of socially interactive events and experiences [19]. It is a feeling that one's peers are having more fun or possess something greater than one is personally experiencing or possessing [20].

When individuals experience greater FoMO, they have been shown to check notifications on mobile devices and social media constantly, which can develop into addictive behaviors related to internet, social media use, and mobile device use [21,22]. Previous literature showed that FoMO can be related to consumer behaviors. For example, Kang and Ma [23] found that people with greater FoMO showed stronger desires to purchase products in order to belong to a particular group (i.e., bandwagon consumption behavior). Similarly, Osemeahon and Agoyi [19] showed that consumers with greater FoMO are more likely to engage in online socially interactive activities, such as being a part of an online brand community in order to build relationships with people who use the same brand. Lee, Sudarshan [7] showed that Gen Z consumers with greater FoMO tend to have stronger beliefs of engaging in shopping behaviors that peers approved of. Through subjective norms, FoMO was shown to be related to purchasing a greater number of apparel products. Applying these findings to the context of social media influencer of the current study, we expect that individuals with greater FoMO may build stronger relationships with social media influencers.

## Psychological, social, and financial wellbeing

We focus on three aspects of wellbeing involving social media: external aspects (social wellbeing), internal aspects (psychological wellbeing), and material aspects (financial wellbeing) [24–26]. Social wellbeing is defined as "the appraisal of one's circumstances and functioning in society" [25, p. 122] and relates to a person's ability to integrate into society with a focus on a sense of belonging and social connectedness [27]. In contrast, psychological wellbeing is a subjective self-evaluation of life related to happiness and satisfaction and focuses more on an internal cognitive evaluation of oneself [24]. Lastly, financial wellbeing consists of several key subjective and objective constructs, such as financial stress, financial security, and financial control [26,28].

The link between social wellbeing needs and meeting these needs through social media use is relatively strong [29]. Individuals try to fulfill their relatedness needs through social media and social networks as it presents a streamlined way to connect with others [30,31]. Further, individuals seek to meet their social wellbeing needs by using social media to pursue a sense of belonging and social support [32]. Social wellbeing needs can often be met online, even if unilaterally, because of the participatory nature that social media provides, specifically through likes and comments [33]. The participatory nature can simulate a "conversation," or the social connectedness that one may be seeking to meet their social needs. This can be especially true if conducted in a "live" social environment in which the user is on social media and interacting with their followers in real-time [34]. However, when social wellbeing needs are not met, social crises, especially feelings of exclusion can emerge. Social media use, while possessing the advantages of social connectedness, can be outweighed by social conflict [30].

Further, an individual's psychological wellbeing can be negatively influenced by social media usage [35]. On social media platforms, people can have control over their self-expression and present themselves in ways that they choose. This allows individuals to publicly, yet selectively, share their lives in an attractive manner that is to be admired by their friends and followers [36,37]. As interpersonal interactions through social media can be shallow and superficial, individuals may compare themselves to the people on social media who present superior and more luxurious lifestyles via upward social comparison [38]. Due to this upward social comparison, research examining social media use has shown that individuals can experience a negative impact to their self-esteem, an internal measure of wellbeing, over time [39]. In addition, this comparison has been related to psychological stress (e.g., depression, anxiety, stress, and loneliness) and lower life satisfactions resulting in lower overall psychological wellbeing [35,39,40]. However, even despite these potential negative implications from social media use, a user's ability to connect live on social media has been shown to mitigate the feelings of loneliness, even if temporarily, thus having the ability to potentially improve an individual's psychological wellbeing [34].

Finally, financial wellbeing can also be influenced by factors related to an individual's social media use. Social media influencers are often self-marketed experts in beauty, fitness, food, and fashion [41] and research has shown that impulsive and compulsive shopping purchases often fall into one of those categories [42]. Social media influencers are experts in compelling consumers to purchase goods impulsively by capitalizing on that individual's materialistic tendencies [43,44]. As a result, impulse purchasing behaviors can lead to financial difficulties, such as consumers overextending their credit [44]. Upward social comparison ideals have also driven consumers to engage in maladaptive shopping behaviors as consumers are purchasing goods to impress, harming their overall financial health [39,43]. Applying these findings to the context of social media influencers of the current study, we expect a strong relationship between PSIs and consumer wellbeing dimensions.

## Social shopping

Social shopping is shopping through social network platforms [45] and combines online shopping with online social networking [46]. Social shopping occurs as followers gain information about products and purchase them via social media influencers, such as using hashtags.

In general, social shopping can shape an individual's sense of community and consumers have been shown to depend on the experiences of a virtual community when making purchasing decisions [47,48]. One place that consumers find these experiences is through social media influencers. Social media fashion influencers are one of the most popular avenues that consumers acquire fashion inspiration from [49]. A growing number of studies have been advancing the understanding of social shopping, specifically examining online shopping behaviors related to consumers being influenced by social media influencers [50]. Social media influencers are able to drive curiosity [50] which is why brands leverage them as strategic marketing instruments to drive social shopping utilization and behaviors [51]. We examine social shopping as a pathway through which PSIs with social media influencer are linked to social, psychological, and financial wellbeing.

## Hypothesis development

### Relationship of FoMO and PSI

Previous researchers in the area of communication technology and mental health have consistently found that individuals with higher levels of FoMO are more likely to have social media addictions [52,53]. Fear and anxiety about missing out on socially interactive

events and experiences has been related to more frequent use of social network [54,55], which can facilitate interactions with social media influencers. Because a PSI is built upon repeated exposure with a persona [15], individuals who use social media more heavily are likely exposed to influencer posts more frequently and may feel that they are friends with the influencer via social media, resulting in stronger PSIs. Social media influencers may be able to exploit an individual's FoMO because they tend to design their posts to be intimate and openly honest with followers [18]. Social media influencers are also considered to be similar to and relatable to followers [51] and are most successful if emphasizing a personal connection with their followers [18]. Consumers who turn to social media influencers to address their FoMO are thus creating an attachment which can grow into a PSI [18,56,57]. Therefore, we propose that individuals who possess a higher level of FoMO build stronger PSIs with social media influencers.

*H1: FoMO is positively related to PSIs with social media influencers.*

## Relationship of FoMO and social, psychological, and financial wellbeing

FoMO has been shown to be negatively related to social wellbeing. Studies show that individuals can develop apprehensions that they are not being accepted by others [58] and become overwhelmed with the concern that they are not as popular as their peers [54,59] as humans have an innate need to belong that can drive much of their behavior [60]. Individuals who experience higher levels of FoMO want to feel as if they belong in their social settings, express higher levels of social envy, and experience feelings that they are being socially excluded [54,61]. Those individuals with greater FoMO also tend to misperceive others as having better lives than themselves and often overestimate other persons' positive experiences while underestimating negative life experiences as compared to their own [61]. Based on these feelings, those with heightened FoMO may evaluate themselves as not being integrated into society well, leading to negative self-evaluations of their social wellbeing. Thus, we propose a negative relationship between FoMO and social wellbeing.

Further, with regard to FoMO and psychological wellbeing, FoMO has been associated with symptoms of depression and anxiety, lower self-esteem, lower body image, and general dissatisfaction with life [30,62]. These results have been identified across several age groups. Specifically, across two studies, one demonstrated that FoMO brought on by social media showed that undergraduate students experienced depression and anxiety symptoms [62] while the other study demonstrated similar anxiety and depressive symptoms, as well as boredom, in adults aged between 18 and 90 [63]. These findings are driven by the evidence that FoMO drive individuals' negative emotions and internal status [8]. Based on the previous literature, we expect a negative relationship between FoMO and psychological wellbeing.

While there was no previous research investigating the direct or indirect relationship of FoMO and financial wellbeing, researchers have shown that FoMO can lead to impulsive purchases [64] and likelihood of purchasing products and services that other people recommended [20]. Further, individuals with greater FoMO tend to purchase products and services that their friends and family recommend in order to belong to the group and to be similar to the majority of people [20,64]. Frequent impulse buying behaviors lead individuals to purchase unnecessary and non-essential products, often leading to credit card debt, which may have a negative impact on their financial wellbeing [64]. Thus, we expect a negative relationship between FoMO and financial wellbeing.

*H2: FoMO is negatively related to (a) social, (b) psychological, and (c) financial wellbeing.*

## Relationship of PSI and social shopping

The growing use of social media has changed how consumers shop online, as well as where they seek product recommendations and who they seek those product recommendations from, as is the case with social shopping [47,65,66]. Social shopping is encouraging consumers to become increasingly dependent on virtual communities to read experiences and reviews of products and promote social media platforms as the perfect place to discover new merchandise [47]. For example, younger generations make fashion and beauty purchases based primarily on what they see on social media, specifically from Instagram posts of influencer accounts [1]. Because influencers cultivate an environment that emphasizes accessibility by appearing to be everyday people, meaning their followers feel they share similar interests, background, and social status, influencers are perceived as relatable and trustworthy [2]. The availability of these trustworthy messages from influencers is often the determining factor in a consumer participating in social shopping activities [47].

Influencers build their brands by sharing valuable content and appealing to a captive audience, which provides a baseline for the PSI to be facilitated [67]. One way their success is measured is through the strength of the PSI, which can be strengthened in two ways: by how they share their content to connect with the audience [16] and by the frequency of content sharing for consumer exposure [68]. Social media posts and advertisements which contain the influencer and the advertised product generate a stronger reaction than when a post only contains the product [16]. This is often due to the established PSI a consumer has with the influencer and not the product. The attention given to the influencer drives mental stimulation which drives the interactions that consumers have with influencers [68] and if consumers are connected to influencers, they are more interested in purchasing influencer-promoted products [16,67]. Thus, we proposed H3 as follows:

*H3: PSIs with social media influencers are positively related to social shopping behaviors.*

## Relationship of PSI with social, psychological, and financial wellbeing

PSIs are likely to come from increased social media use, which can have a positive relationship with social wellbeing if it inflates a person's sense of belonging and feelings of validation [32]. Individuals who rely heavily on PSIs have been shown to translate the feelings achieved from these PSIs to face-to-face interactions [69]. A lot of the actions taken on social media (e.g., liking, commenting, responding to stories/poll) are similar to when interacting with an influencer as with a person whom there is a face-to-face connection [70]. These communication sensations and realizations draw upon the same social skills as a face-to-face relationship that often leads to friendship causing the line between parasocial and truly social to be blurred [71], potentially leading to improved overall social wellbeing. Although PSIs are typically unilateral and lack reciprocity [72], it has been shown that developed PSIs can often replace face-to-face relationships [9]. Therefore, the followers who have a strong PSI with their influencers may be likely satisfied with their social wellbeing through these interactions.

Psychological wellbeing needs, focusing on the inward emotional response, may be met through differently formed PSIs [71]. Individuals are more inclined to form PSI with influencers who actively engage in conversations by responding to comments and messages. This interaction not only makes followers feel valued and appreciated but strengthens their emotional attachment to influencers and enhances their perceptions of authenticity towards influencers [18]. Establishing a genuine and relatable connection akin to friendship with influencers can further enhance overall satisfaction and happiness, contributing to heightened psychological wellbeing in individuals' lives. Moreover, individuals follow influencers

to repeatedly receive updated content that resonates with their interests and aspirations [51], adding enjoyment and excitement into their lives while preventing boredom. Thus, we expect that the PSI with influencers will have a positive relationship with follower's psychological wellbeing by leveraging happiness, enjoyment, and satisfaction in their life.

Social media influencers creating strong PSIs with their followers possess a considerable degree of power over them [73]. This power, promoted through relatable content on social media platforms, has the ability to get consumers to engage in certain acts, such as purchasing items they recommend or partaking in experiences they promote [5,67]. Consumers who are swayed to participate in the actions endorsed by influencers can provoke individuals who try to live a social media worthy lifestyle to go into debt [74,75] and, often times, individuals are swayed to act on impulse when a perceived companion, in this case, an influencer, is involved [76]. Consequently, even the simple act of imagining the presence of an individual, like an influencer, can change self-control habits [77] and potentially prompt impulsive buying behaviors [78], causing a potentially negative outcome on financial wellbeing [79]. Because higher levels of debt have been tied to financial worry and lower overall financial wellbeing [80], we propose a negative relationship between PSI and financial wellbeing.

*H4: PSI with social media influencers is positively related to (a) social and (b) psychological wellbeing while it is negatively related to (c) financial wellbeing.*

## Relationship of social shopping with social, psychological, and financial wellbeing

Social shopping can be a key proponent of social wellbeing as the spirit of social shopping emphasizes interaction and involvement in purchase decisions [81]. Products promoted through online communities often fulfill hedonic motivations [73] and through the purchase of that product, a sense of community can be achieved [82]. Social shopping also often allows individuals to interact with others, providing the opportunity to compare experiences and seek opinions [82]. From buying products that a social media influencer recommended (i.e., social shopping), individuals may have a feeling of being included and integrated into the community that they want to belong, possibly leading to positive self-evaluation about their social wellbeing.

In addition, often times, people make purchases that comply with recommendations found on influencer posts and/or through their social communities. Individuals hope to mimic the levels of attractiveness and prestige through these unidirectional and recommended through social shopping purchases [12]. The decision to engage is often a result of consumers trying to purchase goods that will ultimately move them closer to being their ideal self and the act of buying goods is often an act performed to fulfill psychological benefits, like improving self-esteem [83,84]. Social shopping platforms, like Instagram, Facebook, and TikTok, promote and curate enviable, yet seemingly attainable, lifestyles, and many individuals make having a successful platform to share on career goal for [85]. As a person begins to feel the relatability, they feel that they are able to potentially achieve the lifestyle of an influencer and the follower begins to challenge their own self-efficacy, which is the belief that they should be able to achieve certain behaviors based on perceived similarities [86]. Because of this support, we propose that social shopping will have a positive relationship with individuals' psychological wellbeing.

As consumers typically purchase items via social shopping to belong to their social group and mimic other people's or influencers' behaviors that are shown in social media, unnecessary and unplanned purchase (i.e., impulsive purchasing) may lead to overspending. A line can be drawn from general social media use to financial wellbeing as social shopping

behaviors may influence the economic situation of consumers [73]. If a consumer is over-spending and having greater financial worry due to social shopping, social media use may be negatively related to financial wellbeing [87]. Based on these notions, we propose that social shopping will have a negative influence on financial wellbeing. The research model is shown in Fig 1.

*H5: Social shopping behavior is related to (a) social well-being and (b) psychological wellbeing positively, while (c) financial wellbeing negatively.*

## Methods

### Participants and procedure

After obtaining approval of The Ohio State University Institutional Review Board (IRB Approval No: 2022E0152), data were collected via Amazon Mechanical Turk (Mturk). Mturk workers who lived in the United States were eligible to participate in the survey. In addition, to target the most involved population group regarding social media influencers, three screening questions were included to limit the participants who are aged between 18 and 40, use social media, and follow social media influencers [88]. In particular, we limited to sample to individuals aged 18 to 40, as this group represents young adult consumers who tend to use social media more frequently than older age group [89].

### Measures

The survey questions measured FoMO, PSI, social shopping, social wellbeing, psychological wellbeing, and financial wellbeing. Study variables were measured via well-established, multi-item scales adopted from literature. FoMO was measured with 10-item scale developed by Przybylski, Murayama [8] (e.g., "I fear others have more rewarding experiences than me."). PSI was measured with 12 items adapted from Brown and Bocarnea [90]. Items from the original scale referenced

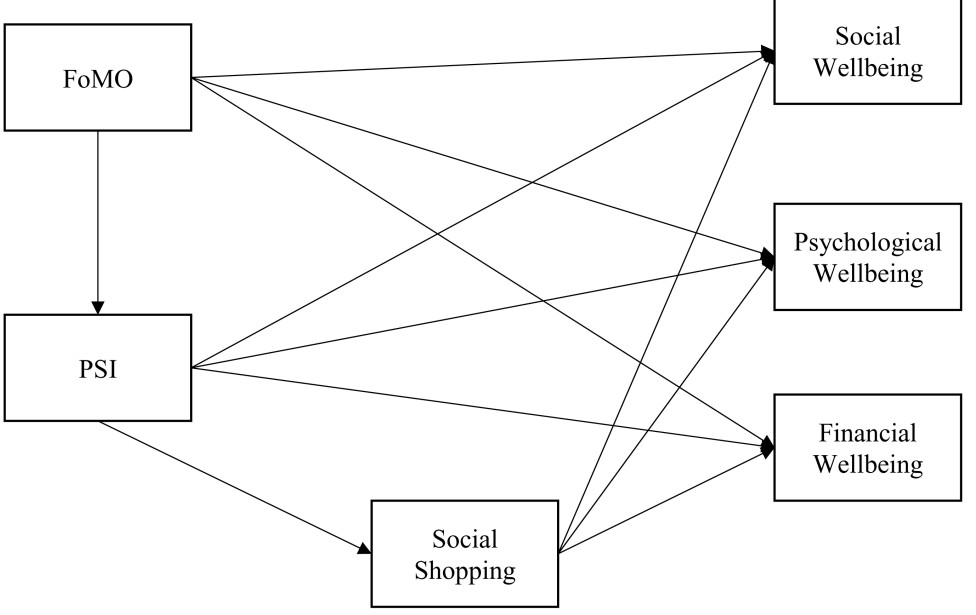

**Fig 1. Research model of the relationships tested in the study.**

television and movie celebrities and were modified to use the term "influencer" in their place. Example item includes "I feel that I understand the emotions of the influencer I follow and share in their experiences." Social shopping was measured with six items; for example, "I am willing to buy items that the social media influencer recommends" [91,92]. Further, social wellbeing was measured with the six items including "It is easy for me to relate to others" [93]. Psychological wellbeing was measured with 12 items, such as "I feel happy with myself as a person" [94]. Lastly, financial wellbeing was measured with 10 items using the Financial Well-Being Scale from the CFPB [87], including "I am just getting by financially." All items were measured using a 5-point Likert scale ranging from 1 = strongly disagree to 5 = strongly agree. Following common practice in reviewing if participants are fully reading and understanding the questions and to minimize careless responses [95–97], three attention check questions were included to measure the quality of responses (e.g., "Based on the responses below, regardless of what your favorite drink is, please select that your favorite drink is coffee. What is your favorite drink?"). The survey questionnaire is shown in S1 Appendix.

## Results

### Descriptive statistics

The initial sample size was calculated by using a rule of thumb calculation of 15 minimum subjects per indicator [98], resulting in 840 subjects for 56 indicators. By taking into account invalid responses, a total of 1,367 responses were collected. The average reported time to take the survey was 28 minutes and 30 seconds. We excluded incomplete responses (n = 287), careless responses that took less than one standard deviation of average time spent (n = 78), and responses where participants answered questions containing simple attention checks incorrectly (n = 139), resulting in 863 valid responses (63% of 1,367).

The average age of participants was 33 years old (*SD* = 7.00). The majority of the respondents identified as male (57%) and selected the married category (65%). Most of the participants were white (74%), had obtained a bachelor's degree (55%), and were employed full-time (85%). The mode income fell between $25,000 and $49,999. Detailed sample characteristics are shown in Table 1.

### Measurement model

To identify measurements composing constructs, we used confirmatory factor analysis (CFA) to assess the measurement model. CFA was performed among six latent variables, including FoMO, social shopping, PSI, social wellbeing, psychological wellbeing, and financial wellbeing. The CFA results are shown in Table 2 and the correlation matrix is shown in Table 3. Due to the low factor loadings below.60, we eliminated two items of the FoMO scale, two items from the social shopping scale, six items from the PSI scale, three items from the social wellbeing scale, one items from the psychological wellbeing scale, and four items from the financial wellbeing scale. After removing these items, the measurement model showed a good fit to the data ($\chi^2$ = 1449.56, $\chi^2$/df = 2.25, CFI =.96, TLI =.95, RMSEA =.03; [99]). All final factor loadings were over.61, all average variance extracted (AVE) values were over.51, and composite reliability (CR) values were over.82, confirming convergent validity [99]. The square root of the AVE for each construct was larger than the corresponding correlation coefficient between the factors, confirming discriminant validity [99].

### Hypothesis testing

We tested all hypotheses using a structural equation modeling (SEM). The SEM result demonstrated an overall good fit to the data: $\chi^2$ = 1482.67, $^2$/df = 2.29, CFI =.95, TLI =.95, and RMSEA =.04 [99]. The results are shown in Fig 2 and Table 4 .

**Table 1. Sample characteristics.**

| Variables | n | % |
|---|---|---|
| Socio-demographic control measures: | | |
| Age | | |
| 18–29 years | 188 | 22% |
| 30–40 years | 665 | 77% |
| No response | 10 | 0.1% |
| Gender | | |
| Male | 489 | 57% |
| Female | 367 | 42% |
| Non-Binary/Non-Confirming, Transgender | 3 | 0.3% |
| No response | 4 | 0.5% |
| Ethnic Background | | |
| Caucasian/White | 640 | 74% |
| African American/Black | 78 | 9% |
| Hispanic or Latino/a/x | 47 | 5% |
| Asian or Pacific Islander | 56 | 6% |
| Native American, Indigenous or Aboriginal People | 20 | 2% |
| Other race/ethnicity not listed, Multiracial | 16 | 2% |
| No response | 6 | 0.7% |
| Marital Status | | |
| Never Married | 259 | 30% |
| Married | 564 | 65% |
| Separated, Divorced, Widowed | 34 | 4% |
| No response | 6 | 0.7% |
| Highest Education Level Completed | | |
| Did not finish H.S. | 1 | 0.1% |
| HS Graduate or GED | 50 | 6% |
| Graduated H.S. but less than 4-year degree | 93 | 11% |
| Associates degree | 50 | 6% |
| Bachelor's degree | 473 | 55% |
| Master's degree | 175 | 20% |
| Professional degree | 6 | 0.7% |
| Doctorate degree | 11 | 1% |
| No response | 4 | 0.5% |
| Employment Status | | |
| Employed, full-time | 733 | 85% |
| Employed, part-time | 67 | 8% |
| Employed, but furloughed | 2 | 0.2% |
| Unemployed, looking for work | 22 | 2% |
| Unemployed, not looking for work | 3 | 0.3% |
| Disabled, unable to work | 4 | 0.5% |
| Retired, Homemaker, Full-time caregiver for family member | 24 | 3% |
| No response | 8 | 0.9% |
| Estimated household pre-tax income, 2021 | | |
| Less than $25,000 | 81 | 9% |
| $25,000 - $49,999 | 276 | 32% |
| $50,000 - $74,999 | 216 | 25% |
| $75,000 - $99,999 | 161 | 19% |

*(Continued)*

**Table 1.** (Continued)

| Variables | *n* | % |
|---|---|---|
| $100,000 - $149,999 | 84 | 10% |
| $150,000 - $199,999 | 23 | 3% |
| $200,000 and above | 11 | 1% |
| No response | 11 | 1% |

*n* = 863.

The SEM results showed a positive relationship between the FoMO and PSI ($\beta$ = .63, $t$ = 13.79, $p$ < .001). This indicates that consumers who experience higher levels of FoMO appeared to have stronger PSIs with social media influencers, supporting H1. Further, we found that FoMO is inversely related to social wellbeing ($\beta$ = -.80, $t$ = −14.00, $p$ < .001), psychological wellbeing ($\beta$ = -.29, $t$ = −6.00, $p$ < .001), and financial wellbeing ($\beta$ = -.67, $t$ = −12.23, $p$ < .001), supporting H2a, H2b, and H2c. The result also showed that the relationship between PSI and social shopping was positive ($\beta$ = .69, $t$ = 14.93, $p$ < .001), which means that consumers who experience PSIs through social media are more likely to engage in social shopping, accepting H3.

PSI was positively associated with all three types of wellbeing: social wellbeing (H4a: $\beta$ = .18, $t$ = 2.96, $p$ < .01), psychological wellbeing (H4b: $\beta$ = .57, $t$ = 8.24, $p$ < .001), and financial wellbeing (H4c: $\beta$ = .15, $t$ = 2.40, $p$ < .05). The result demonstrated that those with stronger PSIs are more likely to have higher levels of social, psychological, and financial wellbeing. As hypothesized, H4a and H4b showed a positive relationship. However, the relationship between PSI and financial wellbeing was negative, which was the opposite direction proposed in H4c. Thus, we accept H4a, H4b, but reject H4c. We did not find a significant relationship between social shopping and three types of wellbeing, rejecting H5a, H5b, and H5c.

This study further examined the mediating roles of PSI and social shopping using the bootstrapping method. Significant indirect effects of FoMO on social shopping ($B$ = .31, $p$ < .01; $B$ is unstandardized coefficient), social wellbeing ($B$ = .14, $p$ < .05), psychological wellbeing ($B$ = .32, $p$ < .01), and financial wellbeing ($B$ = .10, $p$ < .05) through PSI were found. However, the mediating effects involving a single mediator of social shopping and two mediators of PSI followed by social shopping were not found to be significant.

## Discussion

Our results showed that the younger consumer who follows social media influencers and who experiences greater FoMO is more likely to seek a connection, albeit one-sided, to a self-perceived trustworthy online presence. The results confirm for our sample of younger consumers that the anxiety of missing out on socially interactive events and experiences can have individuals turn to social media, which, in turn, can lead to more interactions with social media influencers [54,55]. Since influencers, with whom individuals typically have a PSI, design their content to appear honest and relatable, they are able to cultivate a personal connection with those interact with their content [18,51]. This finding furthers and confirms previous research that indicates that a person experiencing the FoMO may use social media more and seek the pursuit of online "friendships" [55].

In addition, we found that greater FoMO is associated with lower levels of their psychological, social, and financial wellbeing for our sample of younger consumers who follow social media influencers. This finding confirms the small literature that shows that FoMO can be related to negative emotions of anxiety and depression, which are indicators of psychological

**Table 2. Study measures and confirmatory factor analysis results.**

| | Mean (SD) | Factor loading | AVE[a] | CR[b] |
|---|---|---|---|---|
| **FoMO** [8] | 3.17 (1.03) | | .51 | .91 |
| I fear others have more rewarding experiences than me. | | .79 | | |
| I fear my friends have more rewarding experiences than me. | | .82 | | |
| I get worried when I find out my friends are having fun without me. | | .82 | | |
| I get anxious when I don't know what my friends are up to. | | .79 | | |
| Sometimes, I wonder if I spend too much time keeping up with what is going on. | | .71 | | |
| When I go on vacation, I continue to keep tabs on what my friends are doing. | | .70 | | |
| It bothers me when I miss an opportunity to meet up with friends. | | .63 | | |
| When I have a good time it is important for me to share the details online. | | .65 | | |
| **PSI** [90] | 3.56(.84) | | .51 | .86 |
| The influencer I follow makes me feel as if I am with a someone I know well. | | .69 | | |
| I would like to meet the influencer I follow in person. | | .63 | | |
| I feel that I understand the emotions of the influencer I follow and share in their experiences. | | .66 | | |
| I find myself thinking about the influencer I follow on a regular basis. | | .81 | | |
| I have been seeking out information in the media to learn more about the influencer. | | .77 | | |
| I am very much aware of the details of the influencer's life. | | .72 | | |
| **Social Shopping** [91,92] | 3.81(.72) | | .53 | .82 |
| I will consider the shopping experiences of the social media influencer when I want to shop. | | .70 | | |
| I am willing to buy products recommended by the social media influencer. | | .74 | | |
| I would consider buying items that the social media influencer recommends. | | .72 | | |
| I am willing to buy items that the social media influencer recommends. | | .74 | | |
| **Social Wellbeing** [93] | 2.92(1.16) | | .67 | .86 |
| I feel isolated from other people. (R) | | .81 | | |
| When with other people, I felt separate from them. (R) | | .79 | | |
| I feel alone and friendless. (R) | | .85 | | |
| **Psychological Wellbeing** [94] | 3.77(.73) | | .53 | .93 |
| I feel I am able to enjoy life. | | .76 | | |
| I feel I have a purpose in life. | | .72 | | |
| I feel optimistic about the future. | | .72 | | |
| I feel in control of my life. | | .72 | | |
| I feel happy with myself as a person. | | .81 | | |
| I am happy with my looks and appearance. | | .72 | | |
| I feel I am able to live my life the way I want. | | .74 | | |
| I feel confident in my own opinions and beliefs. | | .64 | | |
| I feel able to do the things I choose to do. | | .72 | | |
| I feel able to grow and develop as a person. | | .67 | | |
| I am happy with myself and my achievements. | | .77 | | |
| **Financial Wellbeing** [87] | 2.66(.96) | | .56 | .88 |
| Because of my money situation, I feel like I will never have the things I want in life. (R) | | .78 | | |
| I am just getting by financially. (R) | | .71 | | |
| I am concerned that the money I have or will save won't last. (R) | | .69 | | |
| Giving a gift for a wedding, birthday or other occasion would put a strain on my finances for the month. (R) | | .75 | | |
| I am behind with my finances. (R) | | .79 | | |
| My finances control my life. (R) | | .77 | | |

*Note.*

[a]Average Variance Extracted,

[b]Composite Reliability; (R) Reverse coded.

**Table 3. Correlation matrix.**

|  | FoMO | PSI | SS | SW | PW | FW |
|---|---|---|---|---|---|---|
| FoMO | .74[a] |  |  |  |  |  |
| PSI | .65 | .71[a] |  |  |  |  |
| Social Shopping (SS) | .35 | .69 | .73[a] |  |  |  |
| Social Wellbeing (SW) | .67 | .30 | .12 | .82[a] |  |  |
| Psychological Wellbeing (PW) | .09 | .44 | .37 | .28 | .73[a] |  |
| Financial Wellbeing (FW) | .60 | .33 | .21 | .73 | .17 | .75[a] |

*Note.*

[a]Square root of AVE value for each construct.

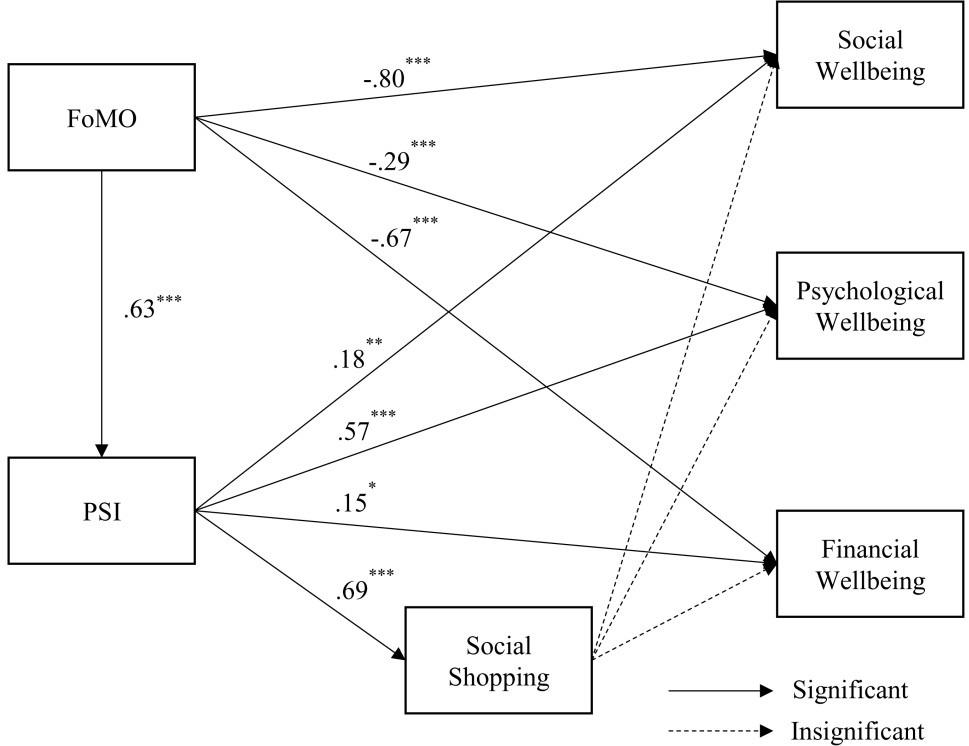

**Fig 2. Structural equation modeling results.** Note: [*]p < .05, [**]p < .01, [***]p < .001. Solid lines indicate significant results, p < .05. Dashed lines indicate insignificant results, p ≥ 0.05.

wellbeing [8]. Similarly, our study shows that a higher level of FoMO is associated with lower levels of social wellbeing for our sample of younger consumers who follow social media influencers. It has been shown that social media influencer-focused consumers struggle in their efforts to belong in their social settings [54,60]. Further, higher levels of FoMO were associated with lower levels of financial wellbeing. This result aligns with previous literature showing that the FoMO is related to impulse purchases [64] and that impulse purchases are related to lower financial wellbeing [28]. Our results document a strong, direct relationship of FoMO and lower financial wellbeing.

Turning to the results for PSI, results showed higher levels of PSI to be associated with more involved social shopping. Social shopping has consumers using virtual communities to make

**Table 4. Structural equation modeling results.**

| Paths | Coefficient |
|---|---|
| *Direct Paths* | β (Standardized coefficient) |
| FoMO → PSI | .63*** |
| FoMO → Social Wellbeing | -.80*** |
| FoMO → Psychological Wellbeing | -.29*** |
| FoMO → Financial Wellbeing | -.67*** |
| PSI → Social Shopping | .69*** |
| PSI → Social Wellbeing | .18** |
| PSI → Psychological Wellbeing | .57*** |
| PSI → Financial Wellbeing | .15* |
| Social Shopping → Social Wellbeing | .05 |
| Social Shopping → Psychological Wellbeing | .08 |
| Social Shopping → Financial Wellbeing | -.07 |
| *Indirect Paths* | B (Unstandardized coefficient) |
| FoMO → PSI → Social Shopping | .31** |
| FoMO → PSI → Social Wellbeing | .14* |
| FoMO → PSI → Psychological Wellbeing | .32** |
| FoMO → PSI → Financial Wellbeing | .10* |
| PSI → Social Shopping → Social Wellbeing | .05 |
| PSI → Social Shopping → Financial Wellbeing | -.05 |
| PSI → Social Shopping → Psychological Wellbeing | .05 |
| FoMO → PSI → Social Shopping → Social Wellbeing | .03 |
| FoMO → PSI → Social Shopping → Financial Wellbeing | -.03 |
| FoMO → PSI → Social Shopping → Psychological Wellbeing | .03 |

Note: $^*p < .05$, $^{**}p < .01$, $^{***}p < .001$.

purchase decisions, as they use the virtual communities to read reviews and seek recommendations [47]. Influencers are using these virtual communities to capitalize on sharing their message and capturing the audience [67]. The current finding aligns with previous research, especially related to influencer marketing, which details that consumers utilize trusted online personas and then engage in social shopping practices [47]. This finding also supports recent marketing initiatives that major retailers have implemented, recognizing that influencers are now acting as retail channels. Firms are completely revising their marketing strategies to include funds budgeted specifically for influencer marketing [5,100]. Social media influencer marketing increased in popularity, about doubling between 2019 to 2024 [101] and in 2025, brands were projected to spend $6.2 billion on influencer marketing in the United States alone [102].

As proposed, individuals who have stronger PSI with influencers are likely to have higher social and psychological wellbeing. PSIs often replace face-to-face relationships [72], meaning psychological and social wellbeing could potentially be achieved through PSIs. PSIs can validate a person's feelings, make them feel as if they belong, and provide a sense of community [103,104]. Further, individuals who have PSIs have been shown to be able to translate feelings derived from a PSI to a face-to-face-relationship, linking it to higher overall social wellbeing [69].

While we hypothesized a negative relationship between PSIs and financial wellbeing, interestingly the results indicate a positive relationship. Financial wellbeing is clearly a subjective measure of how individuals judge their financial situation. Even if influencers are nudging

followers to make purchases of products they recommend, the results indicate that younger consumers who have a stronger bond with social media influencers feel better about their financial situation. Having less financial worries may be an outcome of a boost in self-efficacy from PSIs with social media influencers, as has been shown for non-financial domains [105]. Future research should examine whether the link between PSIs and subjective financial wellbeing is in fact related to self-efficacy in financial matters and whether it is mirrored by objective financial terms.

In addition to the direct effects of FoMO on three types of wellbeing, the findings also revealed a significant mediating effect of PSI on the relationship between FoMO and the three types of wellbeing. Interestingly, while the direct effects of FoMO on the three types of wellbeing exhibited negative relationships, the introduction of PSI as a mediator reversed these associations, rendering the indirect relationships between FoMO and the three types of wellbeing positive. FoMO typically results in negative consequences for individuals' wellbeing, as it often evokes anxieties and concerns about missing out on social interactions and experiences, consequently leading to addictive behaviors related to internet, social media use, and mobile device usage [19,21,22]. However, as demonstrated by the indirect relationship mediated by PSI, individuals experiencing high FoMO tend to develop strong relationships with social media influencers [19], thereby transforming PSI into a channel for fostering positive outcomes regarding individual wellbeing.

Lastly, we did not find a significant between social shopping and social, psychological, and financial wellbeing for our sample of younger consumers who follow social media influencers. The intentions to purchase products recommended by influencers appear to not be associated with self-evaluations of wellbeing. Whether or not younger consumers intend to purchase products recommended by an influencer, they show similar levels of belongingness with a group of people or community (i.e., social wellbeing), evaluations of their life, such as happiness (i.e., psychological wellbeing), and financial situations (i.e., financial wellbeing). These finding may indicate that wellbeing gains of younger consumers stem from the subjective benefits of PSIs rather than the material benefits of the social shopping experience. Alternatively, the relatively low modal income range and low self-reported financial well-being scores in the current sample may explain why social shopping had no significant impact on their well-being measures.

## Limitations and suggestions for future studies

Several limitations should be noted. First, the study is based on self-reported data. While we employed attention checks and excluded careless answers, the survey data may suffer from desirability bias. Second, even though we collected data through MTurk accessing a national sample of consumers, caution needs to be used to generalize the results to U.S. population. In addition, our data is based on participants who lived in the United States. Future research should extend this research to other national and cultural contexts and examine whether our results are consistent across countries. Third, the study data were collected in February 2022, at a time of continued COVID-19 infections. The responses to our focal measures might have been influenced by the exposure to the pandemic. Finally, the study presents a cross-section. Future research should aim to collect repeated measures of younger consumers to disentangle and add time perspective to the role of social media influencer on social, psychological, and financial wellbeing.

## Conclusion

Extant literatures lack sufficient exploration on how FoMO is tied to components of wellbeing through the development of PSIs and social shopping intentions. The present study addresses this research gap by investigating the relationship between a younger consumer's FoMO,

PSIs with influencers on social shopping platforms, and social, psychological, and financial wellbeing. Our results offer two theoretical contributions. First, the study is among the first to investigate the direct relationship of FoMO and PSI. Previous research focusing on PSI identified a number of antecedents, such as homophily and physical attractiveness. Our study is among the first to propose FoMO as a new antecedent and the result document a strong positive relationship between FoMO and stronger bonds with social media influencers.

Second, this study is among the first to demonstrate that the FoMO is directly associated with lower levels of wellbeing across domains, including social, psychological, and financial wellbeing. On the other hand, our findings show that greater FoMO is associated with stronger PSIs, and that through these PSIs, younger consumers have higher levels of social, psychological, and financial wellbeing. The reversal of the association between FoMO and individual wellbeing with the PSI as a mediator highlights the adaptive potential of PSI as a coping mechanism for individuals handling with FoMO. These findings illuminate the intricate interplay between FoMO, PSI, and individual wellbeing, offering valuable theoretical insights into the mechanisms shaping social media dynamics, which has not been explored in the previous literature. This research can further serve as a stepping stone for other researchers investigating the pathways through which the success of influencer marketing can be explained.

Lastly, to our knowledge, this study is among the first to investigate financial wellbeing in the context of social media influencers. We find a positive relationship of PSIs and self-assessed financial wellbeing, pointing to influencers' role for mental stability rather than actual spending – which is also supported by a weak link between social shopping and financial wellbeing for this sample of younger consumers.

## Practical implications

The findings in this study are a platform for understanding how wellbeing is related to mindsets, relationships, and intentions of younger consumers that follow social media influencers. This study provides a better understanding of how the FoMO is related with three facets of wellbeing and the role of PSIs in these relationships.

Findings suggest a positive role of parasocial relationships for social, psychological, and even financial wellbeing for those younger consumers who suffer from higher levels of anxiety and FoMO. Knowledge of this pathway can inform the design of interventions aimed at coping with fears of younger consumers. This study suggests that employing the tool of an approachable, friend-like social media persona can be related to wellbeing outcomes. Our finding that shopping intentions are of lesser importance may point to the potential value of using of social media influencer-based settings when designing interventions for younger consumers.

When taking the business perspective, knowledge of the relationship between PSIs and social shopping suggests that that marketers and businesses can capitalize on this knowledge and design campaigns to improve their sales. Many marketers and businesses already use social media influencers as their marketing tools, and this study further confirms that building a bridge between influencers' posts and retailers to encourage social shopping is a way to increase sales. Retailers may consider using apps to connect influencers' posts to retailers. A growing number of shopping apps helps consumers find the same or similar products that influencers promote [49], tag line "Shop thousands of products tried and styled by real people." [106].

## Supporting information

**S1 Appendix.  Survey questionnaire.**
(DOCX)

## Author contributions

**Conceptualization:** Jung Eun Lee, Cäzilia Loibl.

**Data curation:** Abbey Bartosiak, Jung Eun Lee.

**Formal analysis:** Abbey Bartosiak, Jung Eun Lee.

**Funding acquisition:** Abbey Bartosiak, Cäzilia Loibl.

**Investigation:** Abbey Bartosiak, Jung Eun Lee.

**Methodology:** Abbey Bartosiak, Jung Eun Lee.

**Project administration:** Abbey Bartosiak.

**Software:** Abbey Bartosiak, Jung Eun Lee.

**Supervision:** Jung Eun Lee, Cäzilia Loibl.

**Validation:** Abbey Bartosiak, Jung Eun Lee, Cäzilia Loibl.

**Visualization:** Abbey Bartosiak.

**Writing – original draft:** Abbey Bartosiak, Jung Eun Lee.

**Writing – review & editing:** Abbey Bartosiak, Jung Eun Lee, Cäzilia Loibl.

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
