## [Decision Letter · Decision Letter 0]

29 Dec 2024

PONE-D-24-22539Fear of missing out, social media influencers, and the social, psychological and financial wellbeing of young consumersPLOS ONE

Dear Dr. Loibl,

Thank you for submitting your manuscript to PLOS ONE. After careful consideration, we feel that it has merit but does not fully meet PLOS ONE’s publication criteria as it currently stands. Therefore, we invite you to submit a revised version of the manuscript that addresses the points raised during the review process. With regards to the second reviewer, I agree it would be a good idea to include also descriptive statistics. Don't worry about naming sections, it's really only up to you; they make sense in the current form too.

We look forward to receiving your revised manuscript.

Kind regards,

Frantisek Sudzina

Academic Editor

PLOS ONE

“The authors acknowledge generous funding from the Coca-Cola Critical Difference for Women Grant for Research on Women, Gender, and Gender Equity from The Women’s Place at The Ohio State University.”

“The authors acknowledge generous funding from the Coca-Cola Critical Difference for Women Grant for Research on Women, Gender, and Gender Equity from The Women’s Place at The Ohio State University.”

“The authors acknowledge generous funding from the Coca-Cola Critical Difference for Women Grant for Research on Women, Gender, and Gender Equity from The Women’s Place at The Ohio State University.”

Reviewers' comments:

Reviewer's Responses to Questions

**Comments to the Author**

1. Is the manuscript technically sound, and do the data support the conclusions?

Reviewer #1: Yes

Reviewer #2: Yes

2. Has the statistical analysis been performed appropriately and rigorously? 

Reviewer #1: Yes

Reviewer #2: Yes

3. Have the authors made all data underlying the findings in their manuscript fully available?

Reviewer #1: Yes

Reviewer #2: Yes

4. Is the manuscript presented in an intelligible fashion and written in standard English?

Reviewer #1: Yes

Reviewer #2: Yes

5. Review Comments to the Author

Reviewer #1: Thank you for the opportunity to review this manuscript (PONE-D-24-22539) examining the relationship between Fear of missing out, social media influencers, and the social, psychological and financial wellbeing of young consumers.

A particular strength of the manuscript is its potential to address an important gap in the literature by examining an understudied relationship. As such, the findings have the potential to contribute to the literature on the topic, as well as to the implementation of interventions aimed at informing younger consumers about responsible social media practices. The background and justification for the study is thoroughly discussed and engrained within the academic literature.

However, the manuscript needs some revisions to be published.

1. Actually section 2. “Materials and Methods” should be named “Methods”, with two sub-sections: 2.1. “Participants and Procedure”, describing data collection, ethical standards, and participants; 2.2. “Measures”, more deeply describing the utilized scales;

2. In the section 3. “Results”, author should add a sub-section “Descriptive Statistics and Correlations”, properly describing these lacking data, also to be summarized in two tables;

3. The sub-section 5.2. “Limitations and Suggestions for Future Studies” should be moved at the end of the section 4. “Discussion”, as sub-section 4.1.

Reviewer #2: Figure 1 is hard to read with arrows stacked on top of each other for the three well being boxes. try to move them apart.

"Mturk workers who lived in the United States were eligible to participate in..." - were or were not?

In particular, we limited to sample to those aged 18 to 40 because these ages represent Gen Z and Millennials who use social media more frequently than other generations" - Millennials age range extends beyond 40 in 2022

Considering your sample had relatively low mode income range and score low on the self-reported financial well being measure, these demographic variables might explain why social shopping has no significant impact on any of their well being measures. Consider adding brief explanation of this in conclusion

6. PLOS authors have the option to publish the peer review history of their article (what does this mean? ). If published, this will include your full peer review and any attached files.

**Do you want your identity to be public for this peer review?** For information about this choice, including consent withdrawal, please see our Privacy Policy .

Reviewer #1: **Yes: ** Massimiliano Sommantico

Reviewer #2: No

---

## [Author Response · Author response to Decision Letter 1]

23 Jan 2025

PONE-D-24-22539

Fear of missing out, social media influencers, and the social, psychological, and financial well-being of young consumers

PLOS ONE

Dear Dr. Sudzina and reviewers,

We appreciate the opportunity to revise our manuscript and would like to thank you and the reviewers for your valuable feedback. Below, we provide detailed responses to each comment raised by the reviewers and outline the revisions made to the manuscript accordingly. All changes have been highlighted in the revised manuscript for your convenience.

Editor’s comments:

1. PLOS ONE's style requirements:

Response: We formatted the manuscript and title page to follow PLOS ONE’s style guidelines.

2. Financial Disclosure Statement:

Response: We would like to revise the financial disclosure statement to: "The authors acknowledge generous funding from the Coca-Cola Critical Difference for Women Grant for Research on Women, Gender, and Gender Equity from The Women’s Place at The Ohio State University. The funders had no role in study design, data collection and analysis, decision to publish, or preparation of the manuscript."

3. Acknowledgments Section:

Response: We have removed funding-related text from the Acknowledgments section.

4. Data Availability Statement:

Response: The entire data are freely accessible from the OSF.IO database (URL: https://osf.io/v5zw6/?view_only=3fb68ef318e14d069aa2531de96652f0 ).

5. References:

Response: We have reviewed the reference list for completeness and accuracy and ensured that no retracted articles are cited. Any updates to the references are reflected in the manuscript.

Reviewer #1 Comments:

1. Section 2: "Materials and Methods" should be named "Methods," with two sub-sections: "Participants and Procedure" and "Measures."

Response: We have renamed Section 2 to "Methods" and added two sub-sections including “Participants and Procedure” and “Measures.”

2. In Section 3: "Results," authors should add a sub-section "Descriptive Statistics and Correlations," with data summarized in two tables.

Response: We have added a sub-section titled "Descriptive Statistics" under the "Results" section and Table 1 summarizing the descriptive statistics. Table 3 under the measurement model section presents the correlation matrix.

3.Sub-section 5.2: "Limitations and Suggestions for Future Studies" should be moved to the end of Section 4: "Discussion," as sub-section 4.1.

Response: The "Limitations and Suggestions for Future Studies" section has been relocated to the end of the discussion section.

Reviewer #2 Comments:

1. Figure 1 is hard to read with arrows stacked on top of each other for the three well-being boxes. Try to move them apart.

Response: We have revised Figs 1 and 2 to provide more space between elements, ensuring better readability.

2."Mturk workers who lived in the United States were eligible to participate in..." - were or were not?

Response: We confirm that the original statement is correct: "Mturk workers who lived in the United States were eligible to participate in the survey."

3."In particular, we limited the sample to those aged 18 to 40 because these ages represent Gen Z and Millennials who use social media more frequently than other generations." - Millennials age range extends beyond 40 in 2022

Response: We acknowledge that the Millennials' age range extends slightly beyond 40 in 2022. To address this, we have revised the sentence to: “In particular, we limited to sample to individuals aged 18 to 40, as this group represents young adult consumers who tend to use social media more frequently than older age group.”

4.Considering your sample had relatively low mode income range and score low on the self-reported financial well being measure, these demographic variables might explain why social shopping has no significant impact on any of their well being measures. Consider adding brief explanation of this in conclusion

Response: We added the following statement in the discussion section: “Alternatively, the relatively low modal income range and low self-reported financial well-being scores in the current sample may explain why social shopping had no significant impact on their well-being measures.”

We trust that these revisions address the concerns raised by the reviewers and enhance the quality of our manuscript. Thank you again for your constructive feedback and for considering our work for publication in PLOS ONE. We look forward to your response.

Sincerely,

Cäzilia Loibl

---

## [Editor Report · Decision Letter 1]

27 Jan 2025

Fear of missing out, social media influencers, and the social, psychological and financial wellbeing of young consumers

PONE-D-24-22539R1

Dear Dr. Loibl,

We’re pleased to inform you that your manuscript has been judged scientifically suitable for publication and will be formally accepted for publication once it meets all outstanding technical requirements.

Kind regards,

Frantisek Sudzina

Academic Editor

PLOS ONE
---

## [Editor Report · Acceptance letter]

PONE-D-24-22539R1

PLOS ONE

Dear Dr. Loibl,

I'm pleased to inform you that your manuscript has been deemed suitable for publication in PLOS ONE. Congratulations! Your manuscript is now being handed over to our production team.

Kind regards,

on behalf of

Dr. Frantisek Sudzina

Academic Editor

PLOS ONE